# Cell Growth Inhibition of Saponin XII from *Dipsacus japonicus* Miq. on Acute Myeloid Leukemia Cells

**DOI:** 10.3390/molecules25153325

**Published:** 2020-07-22

**Authors:** Ba Thi Cham, Nguyen Thi Thuy Linh, Do Thi Thao, Nguyen Thi Hoang Anh, Nguyen Thanh Tam, Bui Kim Anh, Isabella Muscari, Sabrina Adorisio, Tran Van Sung, Trinh Thi Thuy, Domenico V. Delfino

**Affiliations:** 1Department of Chemistry, Graduate University of Science and Technology, Vietnam Academy of Science and Technology (VAST), 18 Hoang Quoc Viet, Nghia Do, Cau Giay, Hanoi 100000, Vietnam; BaThiCham@ich.vn (B.T.C.); NguyenThiThuyLinh@ich.vn (N.T.T.L.); DoThiThao@ich.vn (D.T.T.); NguyenThiHoangAnh@ich.vn (N.T.H.A.); NguyenThanhTam@ich.vn (N.T.T.); 2Department of Natural Products Research, Institute of Chemistry, VAST, 18 Hoang Quoc Viet, Nghia Do, Cau Giay, Hanoi 100000, Vietnam; BuiKimAnh2@ich.vn (B.K.A.); TranVanSung@ich.vn (T.V.S.); 3Institute of Biotechnology, VAST, 18 Hoang Quoc Viet, Nghia Do, Cau Giay, Hanoi 100000, Vietnam; 4Section of onco-hematology, Department of Medicine, University of Perugia, 06132 Perugia, Italy; isa.muscari2@gmail.com; 5Foligno Nursing School, Department of Medicine, University of Perugia, 06132 Perugia, Italy; adorisiosabrina@libero.it; 6Section of Pharmacology, Department of Medicine, University of Perugia, Piazzale Severi, S. Andrea delle Fratte, 06132 Perugia, Italy

**Keywords:** *Dipsacus japonicus* Miq., saponin XII, antiproliferative activity, OCI-AML cells, cancer

## Abstract

In previous studies, we isolated the known compound saponin XII from the roots of *Dipsacus japonicus* Miq. Here, we show that this compound reduced the number of acute myeloid leukemia OCI-AML3 cells as evaluated by a hemocytometer. Flow cytometry analyses demonstrated that the reported activity was associated with a significant increase of apoptosis and of cells in the G_0_/G_1_ phase of the cell cycle, with a decrease of cells in the S and G_2_/M phases. Thus, the inhibition of cell growth in OCI-AML3 cells was due to antiproliferative and pro-apoptotic effects. Interestingly, the bioactivity of saponin XII exerted its effect at a concentration as low as 1 µg/mL.

## 1. Introduction

Plants are a valuable source of therapeutically useful compounds, and herbal medicine is the most practiced form of traditional medicine, defined by the World Health Organization (WHO) as “knowledge, skills, and practices based on theories, beliefs, and experiences of an indigenous culture used to maintain health and prevent, diagnose, or treat physical and mental illness” [1]. The value of traditional medicinal plants is not only due to their pharmacological properties but also to the belief that these “natural” remedies are very low in toxicity [1]. However, the beneficial properties of medicinal plants, along with the compounds isolated from them, require globally accepted high standard of manufacturing and biological validation through biotechnological methods [1].

Among many other herbal medicines, the genus *Dipsacus* is widely used in traditional remedies around the world, and is diffused in Europe, Africa and Asia with about 20 species belonging to its family [2]. Different pharmacological properties are attributed to *Dipsacus* roots and seeds. Examples of diseases treated with these plants include lime disease and fibromyalgia, and they are used to purify the liver, kidney, digestive and circulatory systems [2]. Additionally, studies show that compounds isolated from different species of *Dipsacus* exert in vitro cytotoxic activity, and manifest beneficial effects for Alzheimer’s disease, HIV infection, hepatitis B, inflammation and act as pain relievers [2]. *Dipsacus japonicus* Miq. (syn. *Dipsacus lushanensis* C.Y. Cheng and Ai, *Dipsacus tianmuensis* C.Y. Cheng and Z.T. Yin) is a perennial herb, mainly distributed in central and northern China as well as Japan, Korea and Vietnam [3]. It is one of the most popular traditional herbal medicines in Asian countries, especially in China and Vietnam, and is used mainly as a decoction [4,5]. In Vietnam, its dried roots have the effect of eliminating inflammation, relieving pain, treating tendon aches, pimples, sprains and joint pain [4,5]. Members of the *Dipsacus* genus contain a large amount of saponin, which is a glycoside of triterpene. Previous phytochemical studies of *D. japonicus* roots led to the isolation of iridoid and bisiridoid glycosides and hederagenin triterpenoid glycosides [6,7,8,9,10,11]. Recent pharmacological studies demonstrate that the main saponin constituents isolated from the *Dipsacus* genus possess potent cytotoxic activities against tumor cell lines (A549, H157, HepG2 and MCF-7) and several additional promising activities [12,13,14]. The triterpenoid saponins, particularly those isolated from member of the *Dipsacus* genus, have demonstrated various pharmacological activities, such as antinociceptive and cardioprotective cytotoxic activities [12,13,14]. Among saponins, saponin XII was first isolated from *Dipsacus asper* Wall and its biological activities were unclear [15,16]. As part of our chemical study of this plant’s roots, we identified saponin XII (**1**), isolated for the second time in *D. japonicus*, as the second most abundant component [9]. However, the biological activities of this compound were unclear [9]. Since the *Dipsacus* genus exerts potent cytotoxicity (cell death), which acts together with proliferation to define the growth of a cell, we focused on studying the effect of saponin XII (**1**) on cell death, cell cycle and the consequent cell growth as part of a more general testing of medicinal plants on acute myeloid leukemia OCI-AML3 cells [17].

## 2. Results and Discussion

### 2.1. Isolation and Identification of Saponin XII

The chemical structure of purified compound **1** was analyzed by NMR, and further identified by NMR spectroscopy as saponin XII. *Dipsacus japonicus* extraction and separation were modified from a previously described method [9].

HR-ESI-MS of **1** gave a pseudo-molecular sodium adduct ion peak at *m/z* 1567.7136 ([M + Na]^+^, calcd. 1567.7144), corresponding to a molecular formula of C_71_H_116_O_36_ (M = 1544). The ^1^H and ^13^C NMR spectra of **1** (in CD_3_OD) exhibited six methyl singlets, δ_H_ (0.73, 0.83, 0.93, 0.96, 1.00, 1.12, each 3H), and for the analogous carbons at δ_C_ (14.0, 16.1, 17.4, 26.0, 33.0 and 23.6, C24–27; 29–30 respectively). The aglycon was identified as hederagenin by the comparison of the ^1^H, ^13^C NMR and DEPT data obtained in 2D NMR experiments with literature values [13]. The chemical shift of C-3 (δ_C_ 78.6) and upfield chemical shift of C-28 (δ_C_ 176.5) in the ^13^C NMR spectra (in pyridine-*d_5_*) implied that **1** was a bisdesmosidic glycoside, with sugar chains at C-3 and C-28. Sugar aglycon and sugar–sugar linkages were confirmed by the HMBC correlations. Based on spectroscopic analyses, including HR-ESI MS, 1D and 2D NMR experiments, as well as comparison with those reported data, the structure of **1** was determined unambiguously as 3-*O*-[*β*-d-glucopyranosyl(1→4)][*α*-l-rhamnopyranosyl(1→3)]-*β*-d-glucopyranosyl(1→3)-*α*-L-rhamnopyranosyl(1→2)-*α*-arabinopyranosyl hederagenin 28-*O*-*β*-d-glucopyranosyl(1→6)-*β*-d-glucopyranoside (saponin XII), as shown in Figure 1 [9,16].

### 2.2. Biological Activity of Isolated Saponin XII

Saponin XII significantly decreased the number of OCI-AML3 cells at concentrations of 1 and 2 µg/mL (0.648, 1.295 µM) (Figure 2). At these concentrations, the effect of cell number reduction reached its plateau, whereas higher concentrations were toxic to the cells as evidenced by the excessive number of dead cells and by the unreadable cell cycle profile. The decrease in cell number could have been due to apoptotic induction, inhibition of proliferation or both. To evaluate these processes, we stained cell nuclei with propidium iodide (PI) and performed flow cytometry analysis after the exclusion of necrotic cells by forward light scatter (FSC), without the need for annexin V staining [18] to investigate apoptosis and cell cycle of treated vs. untreated cells. As shown in Figure 3, saponin XII (2 and 1 μg/mL) significantly increased apoptotic cell death, although there was no significant difference between these two groups. Since apoptosis is, by definition, a caspase-dependent process [19], experiments to see if the executioner caspase-3 was activated [20] were performed and, as shown, caspase-3 was activated upon treatment with both 1 and 2 μg/mL of saponin XII. In the cell cycle analysis, saponin XII, added for 24 or 48 h, significantly increased the percentage of cells in the G_0_/G_1_ phase and decreased the percentage of cells in the S phase (Figure 4) with an MEC of 1 μg/mL. The reason for the missing analysis after 72 h of treatment with the higher concentration (2 μg/mL) of saponin XII is that no living cells were recovered.

In conclusion, this is the first report on the bioassay of saponin XII. The results indicate that this compound inhibits the cell growth of OCI-AML cells by an increase of apoptosis—a known effect of saponins that is exerted through caspase 3/8/9 activation [13,21]. This effect is shown in particular by *Dipsacus* saponins as previous studies demonstrate that *Dipsacus* saponins decrease apoptosis by increasing the anti-apoptotic molecule Bcl-2, decreasing the expression of pro-apoptotic molecules Bax, caspase-9 and caspase-8 [22], and by decreasing proliferation. Due to a limited amount of the studied compound, it was not possible to perform studies on structure–activity relationships. Hopefully, additional bioactive studies will clarify this interesting effect on acute myeloid leukemia cell lines and examine its potential therapeutic use.

## 3. Materials and Methods

### 3.1. Chemical Methods

*Plant materials:* Roots of *Dipsacus japonicus* (DJ) were bought in the Lan Ong market (traditional herb market) of Hanoi, Vietnam in May 2016. The species was identified by botanist Dr. Nguyen Van Tap, National Institute of Medical Materials, Hanoi. *Method:* Comparison of plant morphology with an authentic voucher specimen. A voucher specimen (Nr. ICH 2016) is deposited in the Institute of Chemistry, VAST, Hanoi.

HR ESI MS spectrum was obtained on a QStar Pulsar (Applied Biosystems). ^1^H NMR (500.13 MHz) and ^13^C NMR (125.77 MHz) spectral data were measured on a Bruker Avance 500 NMR spectrometer at 25 °C. Chemical shifts were expressed in δ (ppm) downfield from as CD_3_OD (^1^H = *δ*_H_ 3.31, 4.78; ^13^C = *δ*_C_ 49.1), and coupling constants were reported in Hertz (Hz). Silica gel 60 F-254 (0.25 mm, Merck); reversed phase RP_18_ F254S (0.25 mm, Merck). CC*:* Silica gel 60 (230–400 mesh, Merck) for the first column, silica gel 60, 40–63 µm (Merck) and Sephadex LH-20 for the following columns. The purity of compound **1** was estimated to be greater than 95% by integration in ^1^H and ^13^C NMR.

### 3.2. Extraction and Purification of Saponin XII

The ground and dried roots of DJ (1 kg) were extracted three times with 90% aqueous MeOH at room temperature. MeOH was evaporated in vacuo and the aq. solution was partitioned with n-hexane followed by EtOAc and *n*-BuOH (each three times), giving three extracts, respectively. The *n*-BuOH extract was separated on silica gel using CHCl_3_-MeOH-H_2_O (65:35:5→60:40:10) to afford 20 fractions (F-1→F-20, 200mL/Fr). The fractions (Fr.16–18) from the above column were combined and further purified by CC on silica gel (CHCl_3_-MeOH-H_2_O, 60:40:10). The main fraction was chromatographed on Sephadex LH-20 using MeOH to provide compound **1**.

### 3.3. Structural Characterizations of Saponin XII *(**1**)*

White amorphous powder; ^1^H NMR (500 MHz, CD_3_OD) δ (ppm) 5.37 (d, *J* = 8.0 Hz, 1H), 5.35 (d, *J* = 1.5 Hz, 1H), 5.27 (t, *J* = 1.5 Hz, 1H), 5.22 (d, *J* = 3.5 Hz, 1H), 4.55–4.53 m, 3H), 4.49 (d, *J* = 8.0 Hz, 1H), 4.37 (d, *J* = 8.0 Hz, 1H), 4.23 (m, 1H), 4.13 (dd, *J* = 11.5, 2.0 Hz, 1H), 4.03 (dd, *J* = 11.5, 2.0 Hz, 1H), 3.95–3.92 (m, 7H), 3.95–3.20 (m, glycoside), 2.88 (dd, *J* = 14.0, 4.5 Hz, 1H), 2.08–1.37 (m, 16 H), 1.31 (d, *J* = 6.0 Hz, 3H), 1.27 (d, *J* = 6.5 Hz, 3H), 1.12 (s, 3H), 1.00 (s, 3H), 0.96 (s, 3H), 0.93 (s, 3H), 0.83 (s, 3H), 0.73 (Appendix A). HR-ESI-MS [M + Na]^+^
*m*/*z* 1567.7136 ([M + Na]^+^, calcd. 1567.7144). Based on ^1^H NMR, ^13^C NMR, HR-ESI-MS, and comparing with literature data [9,16], compound **1** was confirmed to be saponin XII. The purity of saponin XII was determined by HPLC (Appendix A).

### 3.4. OCI-AML Culture Conditions

We used subtype 3 of the OCI-AML cell line, a generous gift from Prof. Maria Paola Martelli (section of hematology, Department of Medicine, University of Perugia). OCI-AML3 cells were maintained in RPMI 1640 medium with 10% fetal bovine serum, 100 U/mL penicillin and 100 μg/mL streptomycin at 37 °C in 5% CO_2_. The cell line was obtained from the German Collection of Microorganisms and Cell Cultures (Braunschweig, Lower Saxony, Germany), kept at logarithmic growth and cultured in 24-well plates to assess the number and morphology of cells. Cultures, kept at 2 × 10^5^ cells/mL, were treated with different concentrations of dimethylsulfoxide (DMSO) (the higher concentration utilized of DMSO was 0.6 μL/mL). The test compounds at the final concentrations of 0, 5, 1 and 2 μg/mL and cells were harvested after 24, 48 or 72 h depending on the different types of experiments.

### 3.5. Analysis of Cell Number, Apoptotic Cell Death and Cell Cycle Progression

Cells were counted manually using a hemocytometer. Cell viability and cell cycle progression were analyzed by flow cytometry to determine the DNA content of cell nuclei stained with propidium iodide (PI) after the exclusion of necrotic cells by forward light scatter (FSC). Briefly, cells were collected by centrifuge and washed in PBS (phosphate-buffered saline). DNA was stained by incubating the cells in PBS containing 50 µg/mL PI for 30 min at 4 °C. This allows direct DNA staining in PI hypotonic solution without the requirement of RNase treatment as the RNA is removed by hypotonic shock [18]. Fluorescence was measured and analyzed by flow cytometry using Coulter Epics XL-MCL equipment (Beckman Coulter Inc., Brea, CA, USA) [23]. Apoptosis data were reported on a logarithmic scale (FL3). For cell cycle analyses, apoptotic cells were gated out and data were reported on a linear scale (FL2). Doublet discrimination was done with the Coulter Epics XL-MCLTM Flow Cytometer SYSTEM IITM Software, which can detect >90% cellular doublet in cells ≥7 μm, using peak vs. integral discrimination. A peak fluorescence signal was assigned to AUX to measure peak vs. integral fluorescence.

### 3.6. Statistical Analysis

Statistical significance was determined using the Mann–Whitney U test as specified in the figure legends. Differences were considered statistically significant according to the following criteria: * *p* < 0.05; ** *p* < 0.01; *** *p* < 0.001.

## Figures and Tables

**Figure 1 molecules-25-03325-f001:**
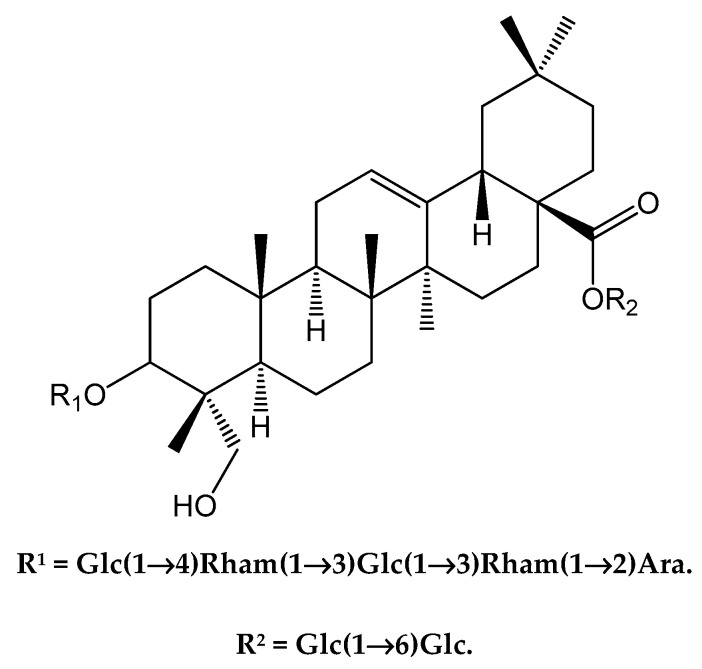
Chemical structure of saponin XII (**1**) isolated from *Dipsacus japonicus* roots.

**Figure 2 molecules-25-03325-f002:**
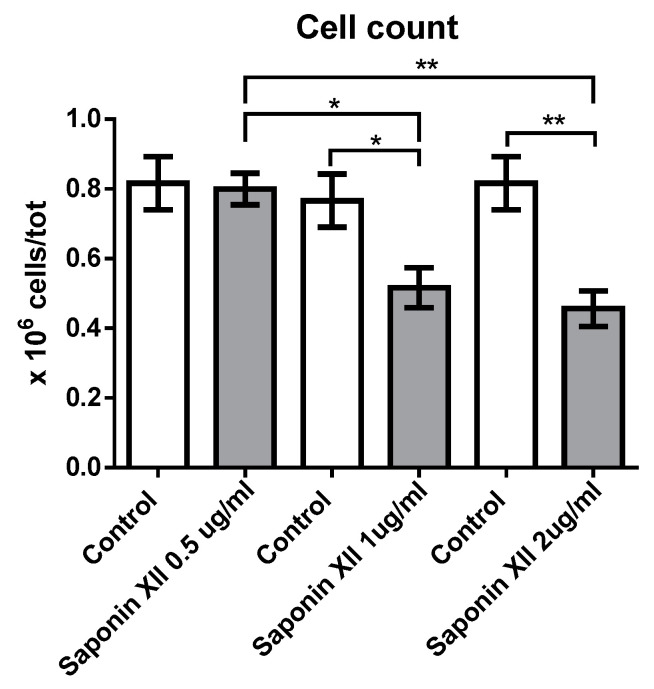
Effects of saponin XII on OCI-AML3 cell number. Bars represent the number of viable cells counted after 24 h of treatment with vehicle (Control) or saponin XII at the concentrations reported on the x-axis. Different amounts of control vehicle (DMSO) were used to dilute different amounts of saponin XII. Data from three independent experiments are reported as mean ± standard error of the mean (SEM). ** *p* < 0.01, * *p* < 0.05.

**Figure 3 molecules-25-03325-f003:**
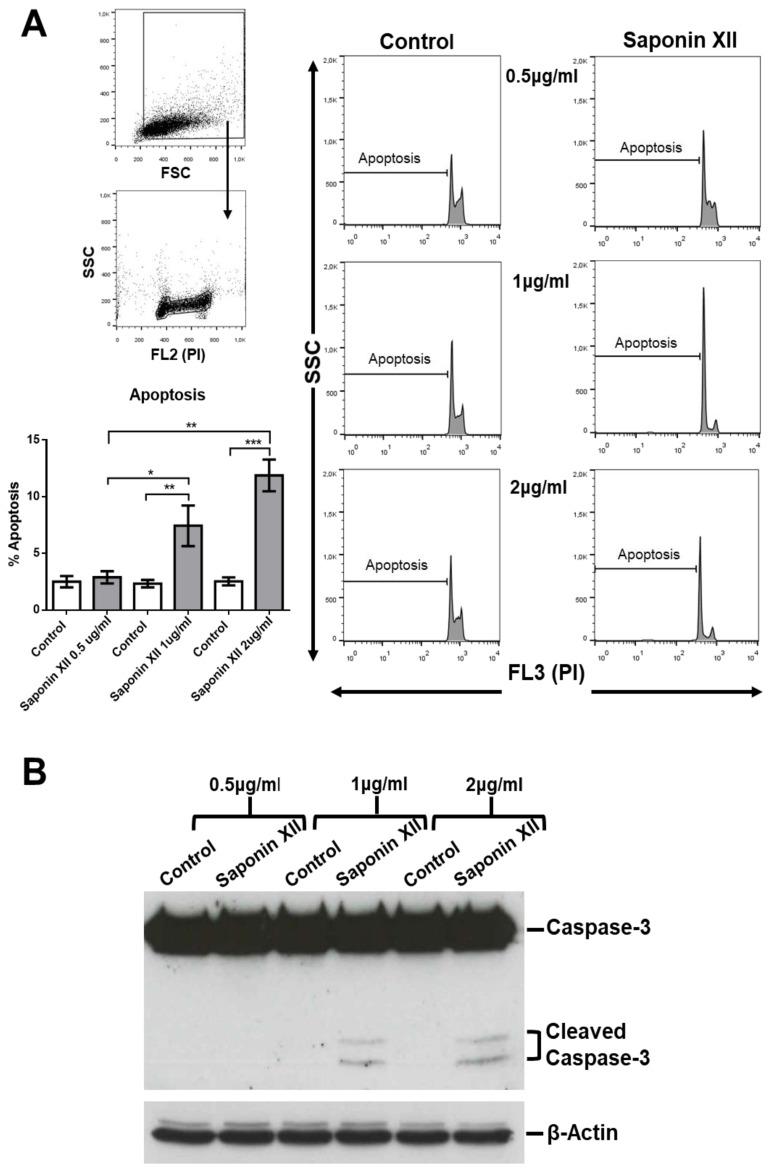
Effects of saponin XII on OCI-AML3 apoptotic cell death. (**A**). In the upper-left panel, the gating strategy to eliminate debris and necrotic cells is shown, and following the arrow, a panel with the gated live cells is shown. Bars represent the percentage of cell death after 24 h of treatment, with control vehicle (Control) or saponin XII at the concentrations reported on the *x*-axis (left side). Histograms from a representative experiment (right side), in which propidium iodide (PI) staining is shown on the x-axes on the logarithmic scale (FL3). (**B**). Western blot analysis of the same groups as in A and probed for activation of caspase-3 (cleaved caspase-3). Data from three (A) or five (B) independent experiments are reported as mean ± SEM. *** *p* < 0.001, ** *p* < 0.01, * *p* < 0.05.

**Figure 4 molecules-25-03325-f004:**
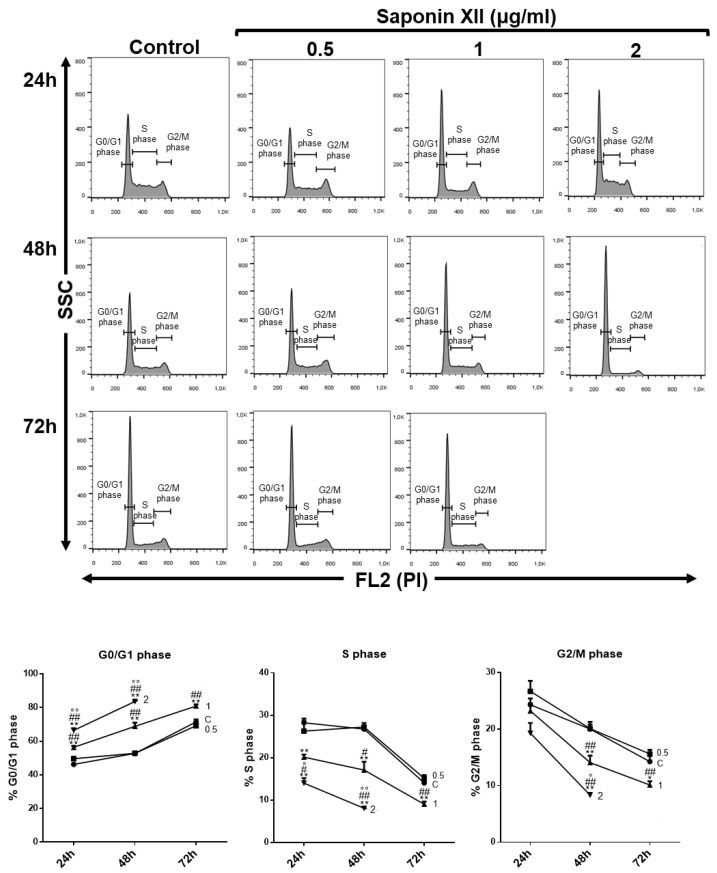
Effects of saponin XII on OCI-AML3 cell cycle progression in time-course experiments. On the upper side, histograms show flow cytometry analyses of representative experiments of cells treated with three different concentrations of saponin XII (0.5, 1 and 2 μg/mL) for 24, 48 or 72 h, in which propidium iodide (PI) staining is shown on the x-axes in linear scale (FL2). On the lower side, lines represent the percentage of cells in the G0/G1 (left panel), S (middle panel) or G2/M (right panel) phase of the cell cycle after 24, 48 and 72 h of treatment with DMSO vehicle (Control) or saponin XII at the concentrations reported. Data from five independent experiments are reported as mean ± SEM. * = significant difference of experimental groups compared to Control; # = significant difference of 1 and 2 compared to 0.5 μg/mL groups; ° = significant difference of 2 compared to 1 μg/mL groups. *^, #, °^
*p* < 0.05, **^, ##, °°^
*p* < 0.01.

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
