# Peer review of "Cell Growth Inhibition of Saponin XII from Dipsacus japonicus Miq. on Acute Myeloid Leukemia Cells"

_molecules, 2020, doi:10.3390/molecules25153325_

Round 1

Reviewer 1 Report

The authors have answered all questions satisfactorily. The addition of Caspase-3 cleavage experiment and timed analysis of cell cycle in presence or absence of sapnonin make the data stronger and more convincing. Good luck.

Minor comments:

#Instead of FL2 or FL3 in Fig 3 and 4 indicate the dyes used.

#3A Indicate live cells within the gate

Author Response

Reviewer 1

Instead of FL2 or FL3 in Fig 3 and 4 indicate the dyes used. 

The dye used was propidium iodide (PI).  This was added in the Fig 3 and 4 and clarified it in the figure’s legends (violet colour).  We did not eliminate FL2 and FL3 because this has been requested from another reviewer.

3A Indicate live cells within the gate

A panel has been added with the indication of live cells.

Reviewer 2 Report

Comments for molecules-872616

The manuscript showed that the compound saponin XII from Dipsacus japonicus Miq. roots can reduce the number of acute myeloid leukemia OCI-AML3 cells as evaluated by hemocytometer. In the revised manuscript, My questions has been responded by authors. Thus I suggest that the manuscript be considered for publication.

Author Response

Reviewer 2

Thank you very much.

Reviewer 3 Report

The revised article by Cham et al analyses the cell growth inhibition of Saponin XII in vitro. The authors have taken reviewer concerns into account and revised the article. That is appreciable!

Two minor concerns are raised.

  1. For the plant voucher, could the authors indicate the identification method used by the expert in Section 3.1? 
  2. For Figure 4 (see Attachment 1), why A* was missing? Please indicate the reason in the Discussion section. And what did the boxes in B* stand for?

Author Response

Reviewer 3

  1. For the plant voucher, could the authors indicate the identification method used by the expert in Section 3.1?

The expert, Dr. Nguyen Van Tap, is a botanist of the “National Institute of Medical Materials” of Hanoi, Vietnam and the method used was a comparison of plant morphology with an authentic voucher specimen.  This has been now added in section 3.1, lines 121-123 (violet colour).

  1. For Figure 4 (see attachment 1), why A* was missing? Please indicate the reason in the Discussion section. And what did the boxes in B* stand for?

A* was missing because the treatment with the higher concentration (2 μg/mL) of saponin XII determined no recovery of cells, so that it was not possible to perform any analysis.  This was already explained in the manuscript (lines 101-102, turquoise colour), but now we changed the sentence to make it more clear (lines 101-102, turquoise and violet colours).

The white boxes in B* are just an editing strategy to show the real lines.  As I wrote above, after 72 hours of treatment with 2 μg/mL of saponin XII, we could not perform any analysis of the cell cycle because we did not recover any live cell.  The graph program used to draw lines (GraphPad) does not allow to make a line with only two time points (24 and 48 hours) in the same graphs in which there are other lines with three time points (24, 48 and 72 hours).  Due to this, we had to draw fake lines (corresponding to the time point of 72 hours) and then to cover them with white boxes.  We now have changed graphical strategy in order to eliminate the visualization of white boxes.

This manuscript is a resubmission of an earlier submission. The following is a list of the peer review reports and author responses from that submission.

Round 1

Reviewer 1 Report

Fig 2 does not show any dose dependent reduction in  cell numbers. Has the cell count been performed manually-this is not mentioned in the methods?

Fig 3 Apoptosis would be better measured using Annexin-PI combination. Gating strategy needs to be shown. PI binds to both RNA and DNA and therefore RNaseA should have been included in the assay.

It would be better of authors can include a few more doses and show dose dependence for apoptosis. It is not possible to verify the conclusions of the authors based on the experiments shown.

Reviewer 2 Report

Comments for Molecules 781134

The manuscript showed that saponin XII from Dipsacus japonicus Miq. can inhibit cell growth of acute myeloid leukemia OCI-AML3 cells. The content of the paper is quite valuable, but there are some problem in experimental design should be solved before the manuscript been considered for publication.

Q1: The result showed that saponin XII inhibits cell growth of acute myeloid leukemia OCI-AML3 cells via the apoptosis pathway. In addition to cell cycle & PI stain, please add discussion of other phenomena concerning saponin XII inhibits cell growth ? For example, discuss cell morphology, and apoptosis markers (Annexin V, caspase-3, 9 and Bcl-2, BAX, DNA fragment) concerning saponin XII inhibits cell growth. Please refer to Li. et al. (2017) Experimental and Therapeutic Medicine 14: 4527-4532.

Q2: In Figure 2 that effects of saponin XII on OCI-AML3 cell number.

  1. Please explain why compare control groups with different concentrations of saponin XII ?
  2. Please add how to calculate the cell number in materials & methods.

Q3: In Figures 3 and 4, please add PI stain with FL-2 or FL-3 parameter in X axis of coordinate graphs.

Reviewer 3 Report

The authors investigated the cell growth inhibition of saponin XII, which was isolated from a traditional herb on acute myeloid leukemia OCI-AML3 cells. The authors reported that Saponin XII decreased the number of OCI-AMIL3 cells via hemocytometer. Additionally, they claimed that the compound increased apoptotic cell death and affect the cell cycle that was examined by the flow cytometry assay. 

Generally speaking, it is a well-written report. However, some minor amendments are needed.

In the introduction, the authors did not provide enough background information about the compound saponin XII. Is it the first time isolated from the herb Dipsacus japonicus Miq.? Is it the primary constituent of the herb/triterpenoid saponins? In line 54, the authors indicated ‘as one of the main components’ that was unclear to me.

Also, it is ambiguous to me why the authors intended to investigate the inhibition of saponin XII on cell growth in acute myeloid leukemia OCI-AML3 cells, as in lines 51 to 53, the authors did not mention the saponins in the herbs could affect cell growth. The authors should clarify the research gap and provide more information in the introduction section. I recommend the authors move lines 76 to 80 back to the introduction. However, I am still confused why the authors chose OCI-AML3 cells, rather than A549, H157, HepG2 and MCF-7, since the main saponins from Dipsacus genus appealed cytotoxic effects to these cell lines (A549, H157, HepG2 and MCF-7) evidenced by previous studies. Or ‘the main saponin constituents’ in line 76 included saponin XII, did it? The authors should provide more justification and make it clear to readers.

For the methods section, Section 3.3, could the authors provide a statement or any references to confirm the compound 1 isolated from the herb by the authors was the actual compound of saponin XII.

For Section 3.6, is it possible to compare the treatment effects (statistical significance) within different concentrations of saponin XII?

In line 102, could the authors provide information relevant to the place of production of the herb bought in the Hanoi market? Also, if the authors could specify which market they bought the herbs from, it should be better.

For Figures 3 and 4, they confused me when I initially looked at the histograms as I could not find any significant differences between the control and saponin XII. Then, I found that their parameters in X-axes were different. Could the authors make sure the parameters in these figures were the same? Then, readers could directly identify the differences between the control and saponin XII groups.

In line 134, please provide the full name of ‘DMSO’ when it first appeared in the manuscript.

In the abstract, line 25, it is inappropriate to use ‘very powerful’ here, because the study only tested the inhibition on one cell line. Moreover, the authors should indicate they only examined one cell line in this study and more acute myeloid leukemia cell lines should be repeated in the future to provide sufficient solid evidence to support the inhibition effects of saponin XII on acute myeloid leukemia cells.

Please add references to the following sentences, including lines 1 to 32 ‘Plants are a valuable source… and mental illness’; lines 32 to 34 ‘ the value of traditional … a very low toxicity’;  lines 37 to 39 ‘Among many other herbal medicines… belonging to its family’; lines 39 to 41 ‘Diseases treated with these plants are… and circulatory systems’; lines 46 to 47 ‘It is one of the most…as a decoction’; and line 49 ‘Dipsacus genus contains… a glycoside of triterpene’. They are not common sense, at least to me. Thus, references are needed.